# The Antineoplastic Effect of Carboplatin Is Potentiated by Combination with Pitavastatin or Metformin in a Chemoresistant High-Grade Serous Carcinoma Cell Line

**DOI:** 10.3390/ijms24010097

**Published:** 2022-12-21

**Authors:** Mariana Nunes, Diana Duarte, Nuno Vale, Sara Ricardo

**Affiliations:** 1Differentiation and Cancer Group, Institute for Research and Innovation in Health (i3S) of the University of Porto, 4200-135 Porto, Portugal; 2Institute of Biomedical Sciences Abel Salazar (ICBAS), University of Porto, 4050-313 Porto, Portugal; 3OncoPharma Research Group, Center for Health Technology and Services Research (CINTESIS), 4200-450 Porto, Portugal; 4Faculty of Pharmacy, University of Porto, 4050-313 Porto, Portugal; 5CINTESIS@RISE, Faculty of Medicine, University of Porto, 4200-319 Porto, Portugal; 6Department of Community Medicine, Health Information and Decision (MEDCIDS), Faculty of Medicine, University of Porto, 4200-450 Porto, Portugal; 7Toxicology Research Unit (TOXRUN), University Institute of Health Sciences, Polytechnic and University Cooperative (CESPU), 4585-116 Gandra, Portugal; 8Department of Pathology, Faculty of Medicine, University of Porto (FMUP), 4200-319 Porto, Portugal

**Keywords:** Carboplatin, chemoresistance, drug repurposing, Pitavastatin, Metformin, high-grade serous carcinoma, synergy

## Abstract

The combination of Carboplatin with Paclitaxel is the mainstay treatment for high-grade serous carcinoma; however, many patients with advanced disease undergo relapse due to chemoresistance. Drug repurposing coupled with a combination of two or more compounds with independent mechanisms of action has the potential to increase the success rate of the antineoplastic treatment. The purpose of this study was to explore whether the combination of Carboplatin with repurposed drugs led to a therapeutic benefit. Hence, we assessed the cytotoxic effects of Carboplatin alone and in combination with several repurposed drugs (Pitavastatin, Metformin, Ivermectin, Itraconazole and Alendronate) in two tumoral models, i.e., Carboplatin (OVCAR8) and Carboplatin-Paclitaxel (OVCAR8 PTX R P) chemoresistant cell lines and in a non-tumoral (HOSE6.3) cell line. Cellular viability was measured using the Presto Blue assay, and the synergistic interactions were evaluated using the Chou–Talalay, Bliss Independence and Highest Single Agent reference models. Combining Carboplatin with Pitavastatin or Metformin displayed the highest cytotoxic effect and the strongest synergism among all combinations for OVCAR8 PTX R P cells, resulting in a chemotherapeutic effect superior to Carboplatin as a single agent. Concerning HOSE6.3 cells, combining Carboplatin with almost all the repurposed drugs demonstrated a safe pharmacological profile. Overall, we propose that Pitavastatin or Metformin could act synergistically in combination with Carboplatin for the management of high-grade serous carcinoma patients with a Carboplatin plus Paclitaxel resistance profile.

## 1. Introduction

There is a long history of drug combinations being widely applied in the treatment of ovarian cancer (OC). The first described efficient combination was Platinum (i.e., Cisplatin or Carboplatin) with Cyclophosphamide [1]. In the 1990s, a trial showed that the combination of Cisplatin with Paclitaxel is more effective compared with the previous regimen [2]. Therefore, Paclitaxel was incorporated into the first-line therapy of OC, which brought significant improvement in the treatment, especially in the advanced disease [2]. Another trial, in OC at stage IIB–IV, demonstrated Carboplatin-Paclitaxel as a good alternative to the Cisplatin-Paclitaxel combination, with similar efficiency and lower toxicity [3].

High-grade serous carcinoma (HGSC) is most often treated with a combination of debulking surgery followed by Platinum-Taxane chemotherapy; still, many patients develop recurrences, harboring cancer cells capable of resistance to first-line chemotherapy [4,5,6,7]. Depending on the mechanisms of action, antineoplastic drugs can be divided into alkylating agents, mitotic spindle inhibitors, antimetabolites, topoisomerase inhibitors, among others [8]. Carboplatin is an alkylating agent applied in the frontline treatment of many neoplasms, including advanced HGSC [9]. This chemotherapeutic compound forms reactive Platinum complexes that cause inter- and intra-strand cross-linkage of DNA molecules, resulting in alterations in DNA structure and inhibition of DNA synthesis, leading to cell death [8,10]. The development of Platinum resistance remains a major obstacle to chemotherapy success as several molecular mechanisms underly this process [8,10,11,12]. Indeed, Carboplatin resistance is a multifactorial process including increased drug detoxification systems, enhanced DNA repair capacity and improved tolerance to nuclear damage, leading to a reduced accumulation of the drug intracellularly and a concomitant decrease apoptosis [8,13,14]. Chemoresistance can occur by intrinsic (innate ability of cancers cells to resist to antineoplastic agents even before being exposed to them) or acquired resistance (tumor present a drug-resistant cell population and a reduced treatment efficacy after chemotherapy exposure), leading to a recurrence capable of evading the cytotoxic effects of chemotherapy [15,16,17]. This is a major problem in oncologic treatment since approximately 80–90% of cancer deaths are associated directly or indirectly with chemoresistance [18,19].

In therapy-resistant tumors, it is essential to search for a pharmacological strategy that could enhance the effectiveness of therapy toward avoiding or reversing chemoresistance [15]. Drug repurposing is a strategy to identify new purposes for approved compounds outside the scope of their original indication, enabling a reduction in the cost of the treatments and the time since the development of new drugs for application in clinical practice [20,21,22]. This concept is centered on the pleiotropic drug effect and on the fact that different diseases can share the same therapeutic targets and molecular features [23]. Moreover, drug repurposing is an easily accessible alternative since they are already available in the pharmaceutical market, presenting pharmacological and toxicological profiles that are well established and their approval for a novel indication can be accelerated compared to the development of new drugs [24,25,26]. Nevertheless, implementing repurposed drugs as an oncological alternative implies the development of adequate clinical trials to evaluate drug efficacy and estimate the maximum tolerated dose to avoid intolerable toxicities [24].

Another approach to overcome chemoresistance comprises the combination of two or more drugs at an optimal synergistic ratio with different mechanisms of action and targeting different pathways, aiming to increase the sensitivity and treatment efficacy [15]. Two or more drugs can network with each other, revealing a synergic, additive, or antagonistic interaction. The synergism is the most desirable drug interaction in the pharmacological context since the combinatory effect of both drugs is much higher than the expected additive effect of each agent [27]. In antagonism, the combined effect of both drugs is less effective than the single activity of each drug [27]. The additivity effect shows that the combination of both agents correspond to the sum of the effects of each compound [27]. Therefore, when two or more drugs combined act synergistically, a better outcome can be achieved as it is possible to use lower doses of each drug, decreasing systemic toxicity and adverse side effects [28,29,30].

In a previous study [31], we confirmed the antitumoral effect of five repurposed drugs — Pitavastatin (antilipidemic), Metformin (antidiabetic), Ivermectin, (antiparasitic), Itraconazole (antifungal) and Alendronate (antiosteoporosis) — in two chemoresistant HGSC cell lines, OVCAR8 (Carboplatin resistant) and OVCAR8 PTX R P (Carboplatin-Paclitaxel resistant) [32]. Interestingly, we have shown that the antitumoral activity of Paclitaxel can effectively be improved when combined with Pitavastatin or Ivermectin, which have an acceptable toxicological profile and can simultaneously increase the activity of Paclitaxel and reduce therapeutical doses [31].

Here, we hypothesized that Pitavastatin, Metformin, Ivermectin, Itraconazole or Alendronate could act synergistically with Carboplatin in two chemoresistant HGSC cell line models (OVCAR8 and OVCAR8 PTX R P). Additionally, we performed these combinations in a normal-like cell line (HOSE6.3) to evaluate the safety pharmacological profile.

Our results demonstrate that the simultaneous combination of Carboplatin with Pitavastatin or Metformin showed the highest cytotoxic effect and the strongest synergism for the Carboplatin plus Paclitaxel resistant cell line (OVCAR8 PTX R P), resulting in an antineoplastic effect superior to Carboplatin alone. Importantly, combining Carboplatin with these two repurposed drugs presented a safe pharmacological profile, not indicating significant effects in the cellular viability reduction in HOSE6.3, a non-tumoral cell line. Overall, our results support a viable therapeutic strategy in the management of HGSC patients with a Carboplatin plus Paclitaxel-resistant profile.

## 2. Results

### 2.1. Repurposing Drugs Increase the Efficacy of Carboplatin in Reducing OVCAR8 and OVCAR8 PTX R P Cellular Viability

Recently, we analyzed the antineoplastic potential of each repurposed drug as a single agent on OVCAR8 and OVCAR8 PTX R P cells using increasing concentrations of each compound, demonstrating that all the drugs have high efficacy in reducing cellular viability for both chemoresistant cell lines [31]. We also showed that all the tested drugs had no or very low efficacy to reduce the cellular viability of HOSE6.3 (normal-like cell line), in contrast to the effects on tumoral cell lines [31]. Overall, these results confirmed that the five repurposed drugs have an acceptable safety profile in normal-like cells, simultaneously presenting significant anticancer efficacy in chemoresistant tumor cells, making them good candidates for being tested in combination with Carboplatin [31]. So, we evaluated the combination of Carboplatin with Pitavastatin, Metformin, Ivermectin, Itraconazole or Alendronate using the combination model previously described [33]. Briefly, chemoresistant HGSC cells were exposed to two drugs alone and combined in a fixed ratio that corresponds to 0.25, 0.5, 1, 2, and 4 fold the individual IC_50_ values of each agent (Figure 1 and Appendix A). Additionally, a morphological evaluation was performed for each treatment condition (Figure 2).

For OVCAR8 and OVCAR8 PTX R P cells, combining Carboplatin with Pitavastatin resulted in a significant increase in anticancer effect (*p* < 0.0001) for 0.25, 0.5 and 1 fold the individual IC_50_ values when compared to Carboplatin as a single agent (Figure 1A,B and Appendix A). The combination of Carboplatin with Metformin caused a significant reduction in cellular viability (*p* < 0.0001) for 0.25, 0.5 and 1 fold the individual IC_50_ values comparing to Carboplatin alone, for OVCAR8 cells (Figure 1C and Appendix A). Moreover, for OVCAR8 PTX R P cells, this combination resulted in a significant increase in antineoplastic efficacy for 0.25 (*p* < 0.005), 0.5, 1 and 2 (*p* < 0.0001) fold the individual IC_50_ values when compared to Carboplatin as a single agent (Figure 1D and Appendix A). For OVCAR8, combining Carboplatin with Ivermectin showed a significant increase in anticancer effect (*p* < 0.0001) for 1 fold the IC_50_ values, compared to Carboplatin as a single agent (Figure 1E and Appendix A). Additionally, for OVCAR8 PTX R P cells, this combination indicates a significant increase in antineoplastic effect (*p* < 0.0001) for 1 and 2 fold the individual IC_50_ values when compared to Carboplatin alone (Figure 1F and Appendix A). The combination of Carboplatin with Itraconazole did not showed a significant reduction in cellular viability compared to Carboplatin alone, for both OVCAR8 and OVCAR8 PTX R P cells (Figure 1G,H and Appendix A). For OVCAR8 and OVCAR8 PTX R P cells, a significant increase in antineoplastic effect (*p* < 0.0001) between Carboplatin with Alendronate and Carboplatin alone was obtained at the concentration of 1 time the individual IC_50_ values (Figure 1I,J and Appendix A). Interestedly, we also showed a significant decrease in cellular viability for 1 fold the individual IC_50_ values when combining Carboplatin with Pitavastatin (*p* < 0.005 for OVCAR8 and *p* < 0.001 for OVCAR8 PTX R P), Metformin (*p* < 0.0001 for both cell lines), Ivermectin (*p* < 0.0001 for OVCAR8 PTX R P) and Alendronate (*p* < 0.001 for both cell lines) when compared to each repurposed drug alone (Figure 1 and Appendix A).

In agreement with the previous results, morphological differences were observed in OVCAR8 and OVCAR8 PTX R P cells for all the combined treatments compared to vehicle and single treatments. The single treatment for each repurposed drugs was published before [31]. The combination of Carboplatin with Pitavastatin, Metformin, Ivermectin and Alendronate at IC_50_ values induced a more aggressive phenotype, i.e., decreasing of cell number, less aggregate formation, and smaller and rounded cells, revealing cell death, when compared to Carboplatin alone (Figure 2). Overall, for these four repurposed drugs tested in combination with Carboplatin, this specific concentration (i.e., IC_50_) demonstrates a significant increase in the cytotoxic effect when compared to Carboplatin or repurposed drugs alone.

### 2.2. Combining Reporposed Drugs with Carboplatin Has a Synergistic Effect on OVCAR8 and OVCAR8 PTX R P Cells

Drug combination is an interesting approach, since this makes it possible to obtain a synergic effect by combining two or more drugs that target different pathways, reducing therapeutic doses, minimizing adverse side effects, and decreasing the capacity of cells acquire multidrug resistance. One of the most acceptable methods to evaluate drug synergism is the Chou–Talalay method due to its quantitative definition, simplicity, flexibility, and efficiency [34,35]. As described by Chou–Talalay, the mass-action law-based determination of synergism is mechanism independent, so this method does not require the knowledge of the mechanisms of action of each drug for the determination of synergism [35]. Considering this, to assess drug synergism, we evaluated the interaction between Carboplatin with the previous repurposed drugs through the Combination Index (CI) obtained using the Chou–Talalay method and plotted on the y-axis as a function of effect level (Fa) on the x-axis (Figure 3). The CI is indicative of synergism (<1), additivity (=1) or antagonism (>1) and Fa is a parameter that varies between 0 (drug does not affect cellular viability) and 1 (drug produces a full effect on decreasing cellular viability) [34,35]. Our results for OVCAR8 cells showed that the combination of Carboplatin with Pitavastatin, Itraconazole and Alendronate did not result in any synergism, presenting a CI > 1 for all the pairs tested (Figure 3A,C). Combining Carboplatin with Metformin resulted in the most promising synergism with two synergic pairs (CI < 1), for 1 and 2 fold the IC_50_ values with a Fa value of 0.914 and 0.933, respectively, for OVCAR8 cells (Figure 3A,C). The combination of Carboplatin with Ivermectin showed the second most promising synergism for OVCAR8 cells with two synergic pairs (CI < 1), producing a Fa value of 0.997 and 1 for 2 and 3 fold the IC_50_ values, respectively (Figure 3A,C). For OVCAR8 PTX R P cells, combining Carboplatin with Pitavastatin showed the most promising synergism, with four pairs being synergic (CI < 1), revealing a Fa value of 0.597, 0.742, 0.872 and 0.978 for 0.5, 1, 2 and 4 fold the IC_50_ values, respectively (Figure 3B,C). Combining Carboplatin with Metformin or Ivermectin resulted in one synergic pair (CI < 1) for 1 and 2 fold the IC_50_ values, respectively, with a Fa value of 0.901 and 0.984 for OVCAR8 PTX R P cells (Figure 3B,C). The combination of Carboplatin with Itraconazole for OVCAR8 PTX R P cells showed two pairs that were synergic (CI < 1), producing a Fa value of 0.096 and 0.967 for 0.5 and 4 fold the IC_50_ values, respectively (Figure 3B,C). The combination of Carboplatin with Alendronate did not result in any synergism for OVCAR8 PTX R P cells presenting a CI > 1 for all the pairs tested (Figure 3B,C). Overall, and in agreement with the previous cellular viability results, in drug synergism evaluation, we observed the same tendency in OVCAR8 and OVCAR8 PTX R P cells for all the combined treatments.

Since the methods used to predict synergism have different mathematical frameworks [36] and can produce slightly different outcomes, we also evaluated the drug interactions using the Bliss Independence and Highest Single Agent (HSA) models (Figure 4 and Figure 5), to compare if the results corroborated the CI values previously obtained by the Chou–Talalay method. The Bliss Independence model stipulates that two compounds produce independent effects, and the predictable combination effect could be assessed centered on the probability of independent events [36,37,38]. The HSA model assumes that the expected combination effect is the highest effect achieved by the most effective drug [36,37,39,40]. For these two models, the synergy score for a drug combination is averaged over all the dose combination measurements giving a positive (synergy, red) and negative (antagonism, green) synergy score values [39,41].

Our results for OVCAR8 cells, by the Bliss Independence and HSA models, revealed a positive synergy score of 1.212 and 3.408, respectively, for combining Carboplatin with Pitavastatin, indicating additivity (Figure 4A,B). Additionally, for both reference models, and concordantly with the Chou–Talalay method, it is possible to note synergic zones (red) for the combinations at lower/intermediate concentrations. In accordance with the results obtained by the Chou–Talalay method, for OVCAR8 cells the combination of Carboplatin with Metformin by the Bliss Independence and HSA models showed a positive synergy score of 10.486 and 15.102, respectively, demonstrating synergism (red) specially for the combinations at lower/intermediate concentrations (Figure 4C,D). For OVCAR8 cells, the Bliss Independence and HSA models indicated that combining Carboplatin with Ivermectin results in a negative synergy score of −5.632 and −3.107, respectively, suggesting additivity (Figure 4E,F). Still, the HSA model showed a synergic zone (red) for the combinations at intermediate/higher concentrations demonstrated similar results with the Chou–Talalay method. The Bliss Independent and HSA models showed for OVCAR8 cells that the combination of Carboplatin and Itraconazole resulted in a negative synergy score of −8.750 and −9.937, respectively, indicating additivity (Figure 4G,H). It is important underline that Itraconazole alone does not have an ideal dose–response curve, so the simulations have a related associated error. For OVCAR8 cells, the Bliss Independence and HSA models revealed that combing Carboplatin and Alendronate results in a synergy score of −3.615 and 1.011, respectively, demonstrating additivity (Figure 4I,J). Still, the HSA model, and concordantly with the Chou–Talalay method, showed a synergic zone (red) for the combinations at intermediate concentrations.

About OVCAR8 PTX R P cells, and according to the Chou–Talalay method, the Bliss Independence and HSA models revealed that combining Carboplatin with Pitavastatin demonstrated a positive synergy score of 6.694 and 14.033, indicating additivity and synergism (red), respectively, predominantly for the combinations at lower/intermediate concentrations (Figure 5A,B). The Bliss Independence and HSA models, in agreement with the Chou–Talalay method, for OVCAR8 PTX R P cells showed that combining Carboplatin with Metformin demonstrated a positive synergy score of 6.396 and 13.868, indicating additivity and synergism (red), respectively, especially for the combinations at intermediate concentrations (Figure 5C,D). The combination of Carboplatin and Ivermectin projected by Bliss Independence and HSA models indicated additivity with a negative synergy score of −7.028 and −1.331, respectively, for OVCAR8 PTX R P cells (Figure 5E,F). Further, according to the Chou–Talalay method, both reference models revealed synergic zones (red), but occur at the intermediate/higher concentrations. The Bliss Independence and HSA models indicated for OVCAR8 PTX R P cells that combining Carboplatin with Itraconazole results in a negative synergy score of −0.433 and −2.141, respectively, suggesting additivity (Figure 5G,H). As we mentioned before, it is important emphasize that Itraconazole alone does not have an ideal dose–response curve. Still, the Bliss Independence model exhibited a synergic zone (red) for the combinations at intermediate concentrations demonstrating similar results with the Chou–Talalay method. The Bliss Independence and HSA models showed that the combination of Carboplatin and Alendronate resulted in a synergy score of −3.797 and 4.882, respectively, demonstrating additivity, for OVCAR8 PTX R P cells (Figure 5I,J). Further, HSA model revealed synergic zones (red), but they occur only at higher concentrations. Moreover, the Bliss Independence model is comparable with the results obtained by the Chou–Talalay method, since presented more antagonistic (green) zones.

As seen in a previous study [31], the results for the synergic effect demonstrate that we can have different scores according to the synergy evaluation models that we choose; however, the three methods assessed demonstrated similar results for all the combinations tested. Therefore, we have shown that combining Carboplatin with Pitavastatin or Metformin are the most promising combination drug pairs for OVCAR8 PTX R P (Carboplatin and Paclitaxel resistant) cell line.

To evaluate the safety pharmacological profile of the interaction between Carboplatin with the repurposed drugs in a non-tumoral cell line (HOSE6.3) we used the previously three reference models. Overall, our results for HOSE6.3 cells by the Chou–Talalay method, reveal that combining Carboplatin with all the repurposed drugs exhibited antagonism with all the five pairs presenting a CI > 1 (Figure 6 and Appendix A). Additionally, and according to the Chou–Talalay method, the Bliss Independence and HSA models revealed that combining Carboplatin with Pitavastatin, Metformin, Ivermectin, Itraconazole and Alendronate displayed a stronger and negative synergy score, indicating antagonism and additivity, according to being <−10 (green) or −10 to 10 (white), respectively (Figure 7).

Overall, the results obtained by the three reference models revealed the same tendency for OVCAR8, OVCAR8 PTX R P, and HOSE6.3 cell lines, demonstrating that all of them generate equivalent outcomes. However, is important to retain that despite the improvements in the reference models [36], they still present some boundaries, as previous discussed [31].

Summing up, our results reveal an acceptable safety pharmacological profile of the tested drugs combinations in a normal-like cell line and reinforce our propose regarding the combination of Carboplatin with Pitavastatin or Metformin in Platinum-Taxane resistant HGSC patients.

## 3. Discussion

Monotherapy schemes highlight alternative molecular pathways in tumor cells, leading to chemoresistance and cancer relapse. The co-administration of two or more drugs enables achieving a better therapeutical effect [42,43], since it is possible to obtain maximum efficacy with the use of lower drug concentrations, decreasing the toxicity and severe side effects [44]. Usually, combining drugs comprises the use of a sensitizing drug and another that increases its cytotoxicity by taking benefit of the vulnerable state of the cells caused by the first agent [43,45]. In many neoplasms, the combinatory regiments are widely applied, being Carboplatin plus Paclitaxel the standard concomitant administration in advanced HGSC management [7,9,46,47]. However, several studies show that, even when combined, the cytotoxic effect remains insufficient, resulting in serious side effects and the emergence of chemoresistant tumor populations limiting the effectiveness of this combination [48,49]. Consequently, alternatives to complement the current Platinum-Taxane-based regimens are needed to decrease the therapeutic dose and exposure time necessary to evade or overcome chemoresistance.

Several studies support that combining chemotherapeutic agents with repurposed drugs increases the therapeutic efficacy by acting through different mechanisms/pathways in a synergistic or, at least, additive way [38,50,51,52]. Carboplatin, a well-recognized alkylating agent, acts by interfering with DNA molecule, generating Platinum–DNA adduct and causing DNA cross-linking, which alters structure and inhibits its synthesis [1,53,54], and, consequently, impairing protein synthesis and cell proliferation [1,54]. Some progress has been made in the search for new drugs that could synergize with Platinum compounds; however, there are a lack of reports using repurposed drugs with Carboplatin for the treatment of HGSC patients. Nagaraj et al. [55] demonstrated that Indomethacin, a non-steroidal anti-inflammatory agent, combined with Cisplatin decreased cell viability more effectively than chemotherapy alone in Cisplatin-sensitive and resistant OC cells, showing an additive outcome. Chloroquine, an anti-malaria drug, in combination with Cisplatin increases the cytotoxicity effect through the induction of lethal DNA damage by reversing Cisplatin resistance in resistant OC cells [56]. Recently, Mariniello et al. [57] demonstrated that Tranilast (antiallergic), Telmisartan (antihypertensive), and Amphotericin B (antifungal) enhanced Cisplatin toxicity via stimulation of Platinum–DNA adduct formation in Platinum-resistant OC cells. Another study showed that Mebendazole, an antiparasitic, acts synergistically by sensitizing chemoresistant OC cells to Cisplatin [58]. Additionally, several reports showed that Disulfiram, an antialcoholism drug, enhances sensitivity to Cisplatin in OC cell lines [59] and in bladder cancer, presenting a synergistic effect [60]. Finally, arsenic compounds, frequently used to treat angiogenic diseases, such as cancer, psoriasis, and rheumatoid arthritis in traditional Chinese medicine, induce cytotoxicity and have a synergistic effect with Cisplatin in Paclitaxel-resistant OC cells [61].

Several preclinical and retrospective studies have demonstrated the potential of using Pitavastatin [62,63,64,65], Metformin [51,66,67,68,69,70], Ivermectin [71,72,73], Itraconazole [74,75,76,77] and Alendronate [78,79,80] as antineoplastic compounds. Here, our purpose was to explore the synergistic effect of co-administrating Carboplatin with these five repurposed drugs in Carboplatin-resistant (OVCAR8) and Carboplatin-Paclitaxel-resistant (OVCAR8 PTX R P) HGSC cell lines. In a previous study, we showed that all these non-oncologic drugs exhibited an antitumoral activity by decreasing cellular viability in a concentration-dependent manner in both chemoresistant cell lines [31]. Here, we tested all five agents in concomitant treatment with Carboplatin, in a combination model previously described by Duarte and Vale [33]. Briefly, both cell lines were exposed to 5 concentrations (0.25, 0.5, 1, 2 and 4 fold the IC_50_ values) of each drug alone and combined with the alkylating agent. Then, the antineoplastic effect of combining Carboplatin with repurposed agents was compared to Carboplatin or each non-oncologic drug as single agents. Our results showed that Pitavastatin and Metformin were the most promising candidates to improve Carboplatin effectivity in OVCAR8 PTX R P cells. Additionally, the morphometry of cells showed consistent effects when compared with the results obtained in the cellular viability assays.

To date, different definitions [52] and methods [38] to assess drug synergism have been described, enhancing the importance of choosing the best reference model according to the data available and study type [36]. The current reference models can be divided into effect-based (e.g., Combination Subthresholding, HSA, Response Additivity, and Bliss Independence models), or dose–effect bases (e.g., Lower Additivity and Zero Interaction Potency), being explained by different mathematical frameworks, based on different definitions of additivity [36]. In this study, drug synergism was assessed by three reference models, namely Chou–Talalay, Bliss Independence and HSA. In drug combination, the Chou–Talalay model postulates a quantitative definition of additive (CI = 1), synergic (CI < 1) and antagonistic (CI > 1) effects based on the median-effect equation, derived from the mass-action law principle and encompassing the Michaelis–Menten, Hill, Henderson–Hasselbach, and Scatchard equations in biochemistry and biophysics [34,35]. On the other hand, the Bliss Independence model defends that two drugs produce independent effects, so the expected combination effect could be calculated centered on the probability of independent events [37,40,45]. Finally, the HSA model stipulates that the expected combination effect is the highest effect achieved by the most effective drug [39,40]. Based on these three reference models, our results for synergy analysis demonstrate that Metformin can synergistically decrease cellular viability for both chemoresistant cell lines. Considering the Chou–Talalay method, our results show more synergistic (CI < 1) pairs for OVCAR8 PTX R P (8/25) when compared to OVCAR8 (4/25) cells. Interestingly, for OVCAR8 PTX R P, Pitavastatin and Metformin are the most promising drugs at lower concentrations. Therefore, the results from the combination of Carboplatin with these two repurposed drugs in OVCAR8 PTX R P are encouraging and suitable for additional experiments to tests its potential effect in patient-derived cells with a chemoresistant disease.

Considering the pharmacotherapy, the best therapeutic agent is the one that selectively destroys neoplastic cells, minimizing adverse side effects in normal cells. In this study, we tested the collateral effects of combining Carboplatin with the five repurposed drugs, and our results showed that the combination therapy has an acceptable safety effect in a non-tumoral cell line.

Statins inhibit 3-Hydroxy-3-Methylglutaryl-CoA Reductase, leading to the blockage of cholesterol biosynthetic pathways [63,65,81]. Previous studies demonstrated that, combining Statins with Cisplatin resulted in an additive or synergic effect in OC cell lines [82,83]. Another study demonstrated that Statin works synergistically with Carboplatin and Paclitaxel [84]. However, these results are not consensual since another study showed an antagonistic effect between Carboplatin and Simvastatin [63]. Nevertheless, Martirosyan et al. [62] showed that Lovastatin sensitizes chemoresistant cells to Doxorubicin by blocking drug efflux pumps. Previously, we showed that OVCAR8 PTX R P cells acquired Paclitaxel resistance by significantly increasing P-glycoprotein (P-gp) expression [32]. In the present work, the results with the double-resistant cell line (OVCAR8 PTX R P), and based on the Chou–Talalay model, showed that Pitavastatin was the most effective repurposed drug considering the lowest IC_50_ values, resulting in four synergic pairs when combined with Carboplatin. On the other hand, for OVCAR8 cells, no synergic pairs were found for the same combination. We hypothesize that the synergic effect of combining Carboplatin with Pitavastatin is related to the chemoresistance background of OVCAR8 PTX R P cell line since are also Paclitaxel-resistant and, therefore, sharing similar mechanism of resistance; however, more studies are required to clarify these results.

Several mechanisms have been suggested for Metformin antineoplastic effects, e.g., modulation of immunological and/or anti-inflammatory responses, inhibition of mTOR, and inhibition of the insulin signals and glucose synthesis via respiratory-chain complex I blockage [81,85,86,87,88]. In endometrioid carcinoma, the use of Metformin at clinically relevant concentrations showed no anticancer activity and combined with Carboplatin did not show synergistic effects [89]. On the contrary, our results with Metformin are in line with many other studies that have showed that Metformin can re-sensitize cells to Cisplatin [67,90,91,92], partially reverse Platinum resistance [93] or have a benefic synergistic effect when combined with Carboplatin [90,94,95]. In the same way, in non-small-cell lung cancer, it was shown that Metformin partially reverses Carboplatin resistance by inhibiting glucose metabolism [96]. Cai et al. [97] showed in breast cancer that the Metformin have a greater efficacy in higher cation transporter-expressing tumors. Patel et al. [98] showed that Metformin combined with Carboplatin or Paclitaxel increases apoptotic activity, implicating a chemo-adjuvant potential in the OC. Nevertheless, our results also indicate that the concentration of Metformin needed to obtain a synergic effect is too high (mM) to be translated to the clinical scenario, therefore, more studies are required to address the clinical significance of these in vitro results.

Many mechanisms can elucidate the antineoplastic effect of Ivermectin, a broad-spectrum antiparasitic agent, such as inhibition of MDR, modulation of Akt/mTOR and Wnt/TCF signaling pathways, and inactivation of PAK-1 expression [99,100,101,102,103,104,105]. Kodama et al. [71] showed that the combination of Paclitaxel with Ivermectin produces a stronger antitumoral effect on OC than each drug alone. Additionally, it was shown that Ivermectin augments Cisplatin efficacy in OC by suppressing the phosphorylation of key molecules in Akt/mTOR signaling [72]. In addition, this antiparasitic drug has been reported to be capable of reversing the chemoresistance in colorectal, breast, and chronic myeloid leukemia cancer cells by inhibiting the EGFR/ERK/Akt/NF-κB pathway [73]. Recently, Juarez et al. [106] demonstrated that Ivermectin was synergistic with Docetaxel, Cyclophosphamide and Tamoxifen in breast and prostate cancer cell lines. These studies corroborate our results regarding combining Carboplatin with Ivermectin, where we observed a significant increase in the antineoplastic effect for chemoresistant cell lines, with a synergistic, or at least, an additive effect.

Itraconazole, a common antifungal agent, has a potential anticancer effect when used in clinically recommended doses [107]. Many studies indicate that Itraconazole targets different mechanisms, including reversing chemoresistance mediated by P-gp, inhibiting Hedgehog, mTOR, and Wnt/β-catenin signaling pathways, and reducing angiogenesis and lymphangiogenesis [107,108,109,110,111,112]. *In vitro* studies indicate that Itraconazole can reverse P-gp-mediated resistance associated with Docetaxel, Paclitaxel, Vinblastine, Daunorubicin, and Doxorubicin in a concentration-dependent manner [74,77,113]. More recently, several preclinical and clinical trials indicate that Itraconazole can reverse Paclitaxel resistance [87,111,112]. Another study using patient-derived xenografts models derived from OC chemoresistant patients showed a synergistic effect of combining Itraconazole with Paclitaxel [76]. Despite these interesting findings, our combination results of this antifungal agent did not lead to an accurate IC_50_ value, making it impossible to infer its therapeutic effect in our cell line models.

Bisphosphonates block farnesyl pyrophosphate synthase, located downstream HMGCR, leading to the impairment of cholesterol biosynthesis [114]. Preclinical evidence proposes that Bisphosphonates have an antineoplastic activity [115,116,117,118,119]. Bisphosphonates present an antitumoral property when combined with chemotherapeutic agents inhibiting tumor proliferation and dissemination of OC [91,92,114]. Combining Zoledronic Acid with Cisplatin, Doxorubicin or Paclitaxel showed a synergistic effect by increasing the cell death in breast cancer cells [120,121,122]. Göbel et al. [123] also showed a promising antitumor effect of Zoledronic Acid in the OC. Another study revealed that Alendronate decreases stromal invasion, tumor burden, and ascites, suggesting an antineoplastic activity in OC [124]. Knight et al. [125] demonstrated a direct effect of Bisphosphonates in OC cell lines and tumor-derived cells; however, when combining with Cisplatin or Paclitaxel, the combination did not result in a significant increase its chemotherapeutic effect. Our results are in agreement to this last study, since we observed an effect of Alendronate in both chemoresistant cell lines [31], but when this agent was combined with Carboplatin we observed an antagonistic or additive effect, not being the best drug partner to combine with this alkylating agent.

The Platinum resistance mechanisms are extremely complex since a lot of cellular events have been described, e.g., changed cellular drug accumulation, increased detoxification systems, improved DNA repair process, and decreased apoptosis and autophagy [126,127]. The uptake of Platinum-based antineoplastic agents is mediated by multiple transporters; however, the altered expression level, localization, or activity may decrease the intracellular accumulation of Platinum compounds and consequentially diminish their cytotoxic effect [126]. Moreover, detoxification systems can bind to these compounds and prevent the formation of Platinum–DNA adducts [126]. Additionally, the activation of the DNA repair process, an increase in apoptosis or autophagy can deeply influence the Platinum response [126].

One of the most important hallmarks of resistance in Platinum-resistant cell lines is the reduced cellular content [128]. To ensure the cytotoxic effect is necessary to guarantee the accumulation of Platinum compounds inside tumoral cells, and an amplified cellular efflux or diminished cellular influx is associated with chemoresistance since it prevents the exposure of cancer cells to lethal concentrations of the drugs [126]. Facing this scenario, it is crucial to identify the transporters responsible for Platinum uptake and intake and find effective ways to target them to evade or reverse chemoresistance. An interesting study demonstrated that an improved expression of efflux transporters that mediate copper homeostasis, i.e., P-type ATPases (ATP7A and ATP7B), predicts shorter survival for OC patients treated with Platinum compounds and demonstrate the important role of these transporters in chemoresistance [129]. In Platinum resistance cells, these transports are altered in its cellular localization contributing to drug retention [130,131]. So, downregulation of ATP7A/B could be an effective way to overcome chemoresistance [126]. The Copper transporter 1 (CTR1) is a transporter of Carboplatin, and many studies indicate that low expression of their reduces the intracellular accumulation of Platinum compounds [132], while high expression sensitizes cancer cells to these agents [133,134]. More studies are needed to confirm the results shown here and it is crucial to assess if Pitavastatin and Metformin act as chemosensitizers by being substrates and modulators of these (or other) important proteins, inhibiting their function and, consequentially, enhancing the cytotoxic efficacy of the antineoplastic drugs. The Platinum resistance is multifactorial and disclosing the molecular mechanisms of resistance and identifying good molecular predictive biomarkers of Carboplatin resistance are yet to be discovered.

## 4. Materials and Methods

### 4.1. Cell Lines and Culture Conditions

The OVCAR8 (Carboplatin-resistant) [50] and OVCAR8 PTX R P (Carboplatin and Paclitaxel-resistant) [32] cell lines were selected as HGSC models. OVCAR8 was kindly provided by Doctor Francis Jacob, Gynecological Cancer Center and Ovarian Cancer Research, Department of Biomedicine, University Hospital Basel and University of Basel, Basel, Switzerland. OVCAR8 PTX R P was established in our laboratory as previously described [32]. Additional experiments were carried out in a non-tumoral cell line (HOSE6.3), described as a human ovarian epithelial cell line established from a normal ovary, surgically removed from patients with non-malignant disease [135]. Cells were grown in RPMI-1640 medium, GlutaMAX^TM^ Supplement, HEPES (ThermoFisher Scientific, Waltham, MA, USA), supplemented with 10% (*v*/*v*) inactivated and filtered fetal bovine serum (Biowest, Nuaillé, France) and 1% (*v*/*v*) penicillin/streptomycin (ThermoFisher Scientific, Waltham, MA, USA) and maintained at 37 °C and 5% CO_2_. All the cell lines were authenticated using short tandem repeat profiling and regularly tested for the absence of mycoplasma.

### 4.2. Drugs

Carboplatin, Pitavastatin, Metformin, Ivermectin, Itraconazole and Alendronate were purchased from Selleckchem (Houston, TX, USA), dissolved in dimethyl sulfoxide (DMSO; AppliChem, Barcelona, Spain) or distilled water and stored at −80 °C, according to the manufacturer’s instructions. Immediately prior to use, an aliquot was diluted at the required concentrations.

### 4.3. Cell Viability Assay

To determine the effect of single and combination drug treatments on the cellular viability, a resazurin-based assay—Presto Blue—was performed as described before [31].

### 4.4. Drug Treatment and Interaction Analysis

The IC_50_ values for Carboplatin and each drug repurposed drug alone were obtained in previous reports [31,32] and were used for the combination studies that were performed according to the previously described method [39], using increasing concentrations of both drugs in a fixed ratio, as suggested by Chou–Talalay [136]. Briefly, Carboplatin was combined in a simultaneous treatment with different repurposed drugs in fixed-dose ratio that corresponds to 0.25, 0.5, 1, 2, and 4 fold the individual IC_50_ values for 48 h.

To measure drug interactions between Carboplatin and repurposed drugs, we calculated the CI by the Chou–Talalay method [35] using the CompuSyn Software v1 (ComboSyn, Inc., New York, NY, USA). A mutually exclusive model, assuming that drugs act through entirely different mechanisms, was used for this analysis [137]. The CI is a quantitative representation of pharmacological interactions (CI < 1, synergism; CI = 1, additivity; and CI > 1, antagonism), plotted on the y-axis as a function of Fa on the x-axis to assess drug synergism between drug combinations. Additionally, we estimated the expected drug combination responses based on the Bliss Independence and HSA reference models using SynergyFinder 2.0 Software (Netphar, Faculty of Medicine, University of Helsinki, Helsinki, Finland) that allow an interactive analysis and visualization of multidrug combination profiling data [37,45]. The synergy score for a drug combination is averaged over all the dose combination measurements giving a positive and negative synergy score values could be observed in 2D and 3D synergy maps dose regions and denote synergy (red) and antagonism (green), respectively [39,41]. The cNMF algorithm implemented in SynergyFinder 2.0 Software was used for estimation of outlier measurements [138].

### 4.5. Microscopic Evaluation

All microscopic figures were obtained under a Leica DMi1 inverted phase contrast microscope (Leica Microsystems, Wetzlar, Germany), at 50× magnification.

### 4.6. Statistical Analysis

All assays were performed in triplicate with at least three independent experiments. Data were expressed as the mean ± standard deviation (SD), and statistical analysis was carried out in GraphPad Prism 8 (GraphPad Software Inc., San Diego, CA, USA) using ordinary one-way or two-way ANOVA followed by Šıdák’s multiple comparison test.

## 5. Conclusions

Combining Carboplatin with Pitavastatin or Metformin may be a promising therapeutic approach for Carboplatin plus Paclitaxel chemoresistant HGSC patients since both drugs present distinct mechanisms of action, suppressing different chemoresistance mechanisms/pathways. To the best of our knowledge, our study is the first in showing that Pitavastatin and Metformin alone and combined with Carboplatin increase the cytotoxic effect of this antineoplastic drug in a Carboplatin plus Paclitaxel chemoresistant HGSC model.

In pharmacological studies, the most important results are the effects of combining two or more drugs at lower concentrations. Nevertheless, we consider important deeper mechanistic studies to understand the anticancer effect of combining Carboplatin with Pitavastatin or Metformin in Carboplatin-Paclitaxel chemoresistant HGSC patients. The next step is evaluating the effect of these combinations in a panel of cell lines with different treatment backgrounds and expand our research to *ex vivo* models, since different patients present specific phenotypic characteristics, genotypic status and chemoresistance patterns. Indeed, the use of patient-derived organoids will allow the development of drug sensitivity test to predict clinical responses to therapy and identify more efficient regiments.

Overall, our results highlighted Pitavastatin and Metformin as chemosensitizing agents of Platinum-taxane resistance, supporting further research.

## Figures and Tables

**Figure 1 ijms-24-00097-f001:**
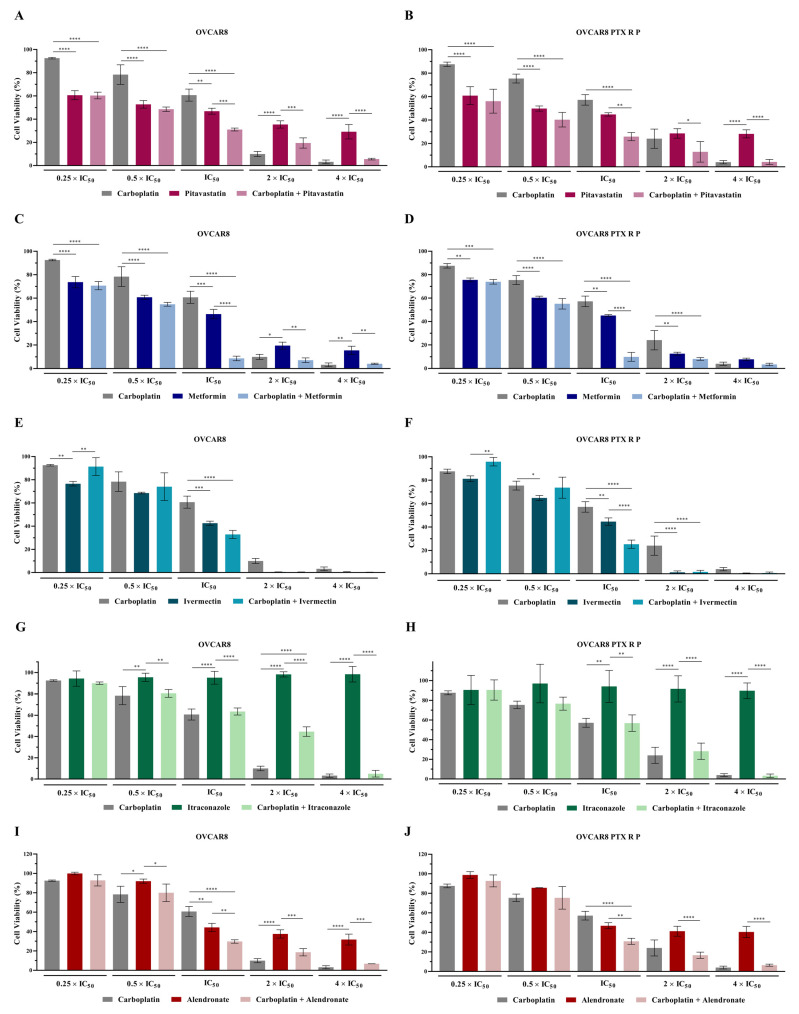
Repurposed drugs increase the efficacy of Carboplatin in reducing cellular viability of chemoresistant high-grade serous carcinoma cells. (**A**–**J**) Bar charts showing cell viability of OVCAR8 (**left**) and OVCAR8 PTX R P (**right**) cells obtained by the Presto Blue assay after exposure to fixed dose ratios that correspond to 0.25, 0.5, 1, 2 and 4 fold the individual IC_50_ values of each drug, e.g., Carboplatin combined with (**A**,**B**) Pitavastatin, (**C**,**D**) Metformin, (**E**,**F**) Ivermectin, (**G**,**H**) Itraconazole and (**I**,**J**) Alendronate for 48 h. The combined treatment was co-administered at the same time. All assays were performed in triplicate in at least three independent experiments. Data are expressed as the mean ± standard deviation and plotted using GraphPad Prism Software Inc. v6 (GraphPad Software Inc., San Diego, CA, USA). Statistical analysis was performed using ordinary one-way ANOVA followed by Šıdák’s multiple comparison test (**A**–**J**) and values of * *p* < 0.05, ** *p* < 0.001, *** *p* < 0.005, and **** *p* < 0.0001 were considered statistically significant.

**Figure 2 ijms-24-00097-f002:**
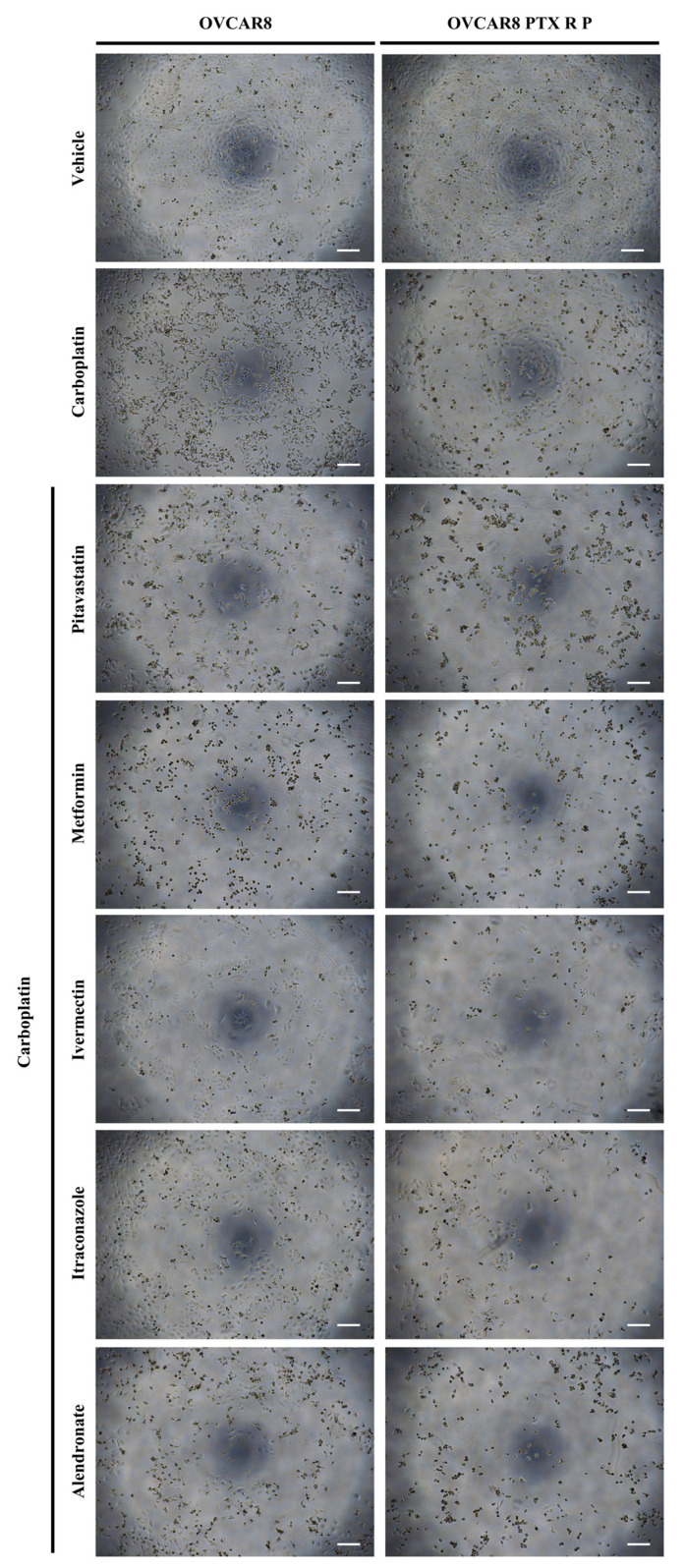
Repurposed drugs improved the cytotoxic effect of Carboplatin on chemoresistant high-grade serous carcinoma cells. Representative microscopy images of OVCAR8 and OVCAR8 PTX R P cells after exposure to vehicle, Carboplatin, Carboplatin + Pitavastatin, Carboplatin + Metformin, Carboplatin + Ivermectin, Carboplatin + Itraconazole and Carboplatin + Alendronate at IC_50_ values of each drug for 48 h. All assays were performed in triplicate in at least three independent experiments. Scale bar, 200 μm.

**Figure 3 ijms-24-00097-f003:**
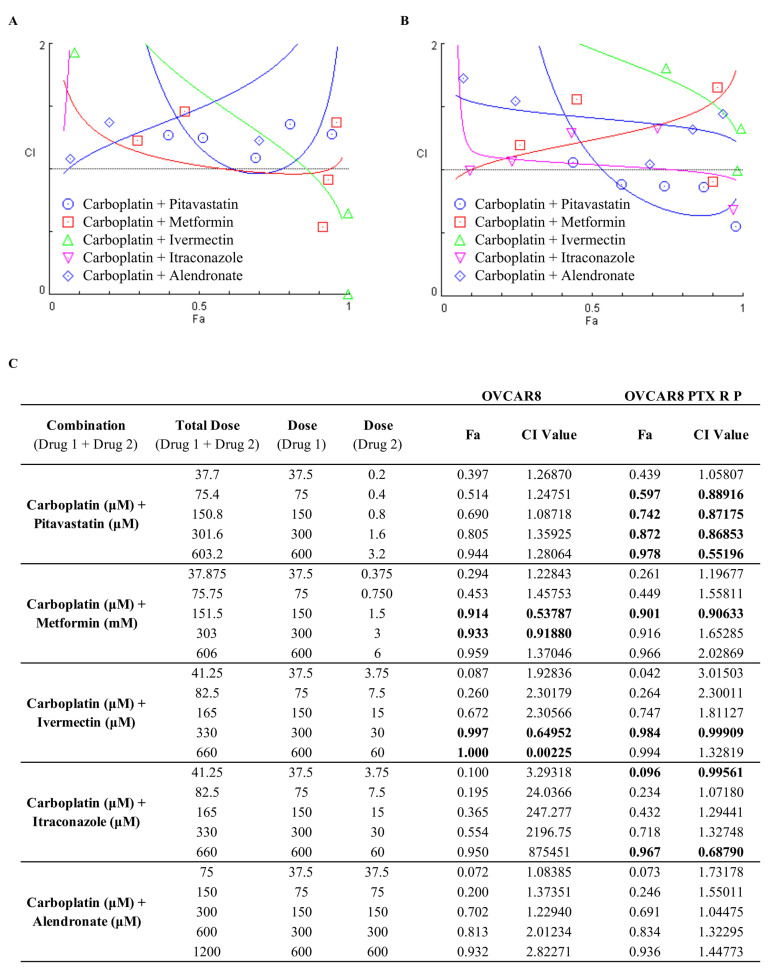
Combining Carboplatin with repurposed drugs has a synergistic effect on chemoresistant high-grade serous carcinoma cells. (**A**,**B**) Chou–Talalay method effect level (Fa)—Combinatory Index (CI) plot and (**C**) Fa values and respective combinatory index (CI) values showing drug synergism of (**A**) OVCAR8 and (**B**) OVCAR8 PTX R P cells, after exposure to fixed dose ratios that correspond to 0.25, 0.5, 1, 2 and 4 fold the individual IC_50_ values each drug, e.g., Carboplatin combined with Pitavastatin, Metformin, Ivermectin, Itraconazole and Alendronate for 48 h. The combined treatment was co-administered at the same time. All assays were performed in triplicate in at least three independent experiments. CI was plotted on the y-axis as a function of Fa on the x-axis to evaluate drug synergism. CI: <1 (synergism, bold), =1 (additivity) and >1 (antagonism).

**Figure 4 ijms-24-00097-f004:**
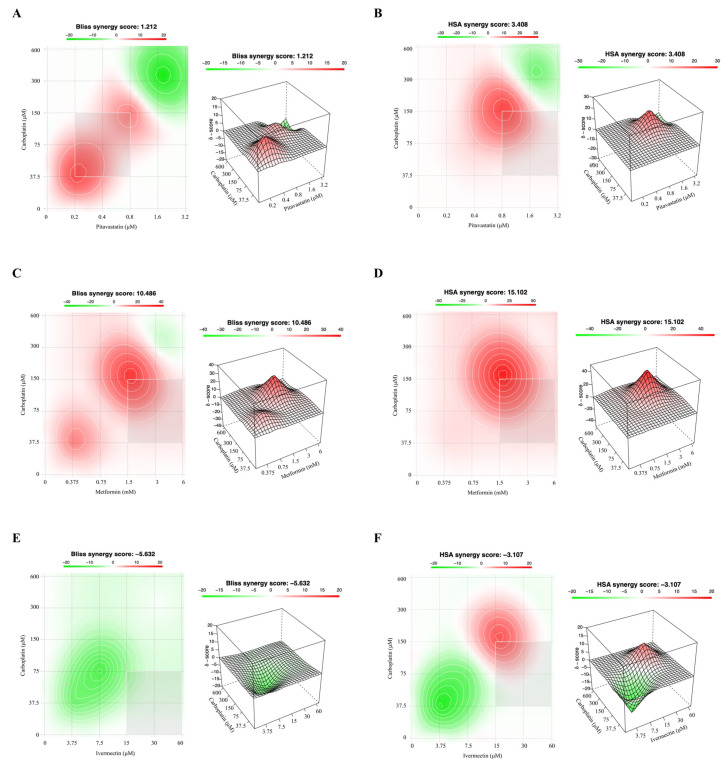
Combining Carboplatin with Metformin has a synergistic effect on OVCAR8 cells. (**A**–**J**) Bliss Independence and High Single Agent (HSA) synergy 2D and 3D plots showing drug synergism of OVCAR8 cells, after exposure to fixed dose ratios that correspond to 0.25, 0.5, 1, 2 and 4 fold the individual IC_50_ values each drug, e.g., Carboplatin combined with (**A**,**B**) Pitavastatin, (**C**,**D**) Metformin, (**E**,**F**) Ivermectin, (**G**,**H**) Itraconazole and (**I**,**J**) Alendronate for 48 h. The combined treatment was co-administered at the same time. All assays were performed in triplicate in at least three independent experiments. Synergy score: <−10 (antagonism, green), −10 to 10 (additivity, white) and >10 (synergism, red).

**Figure 5 ijms-24-00097-f005:**
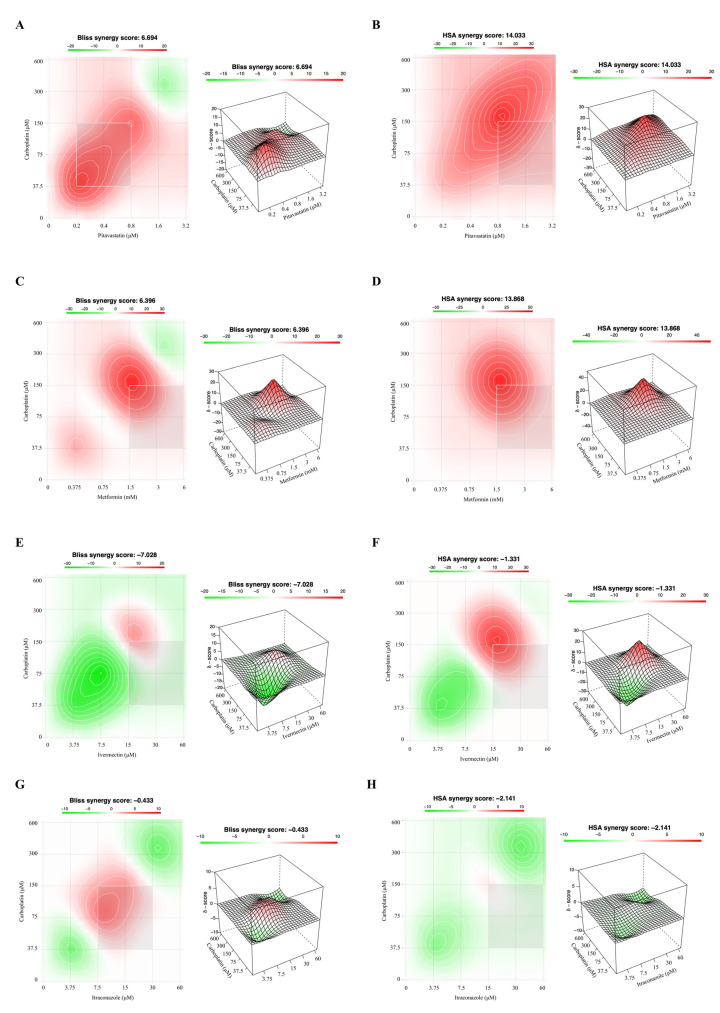
Combining Carboplatin with Pitavastatin or Metformin has a synergistic effect on OVCAR8 PTX R P cells. (**A**–**J**) Bliss Independence and High Single Agent (HSA) synergy 2D and 3D plots showing drug synergism of OVCAR8 PTX R P cells, after exposure to fixed dose ratios that correspond to 0.25, 0.5, 1, 2 and 4 fold the individual IC_50_ values each drug, e.g., Carboplatin combined with (**A**,**B**) Pitavastatin, (**C**,**D**) Metformin, (**E**,**F**) Ivermectin, (**G**,**H**) Itraconazole and (**I**,**J**) Alendronate for 48 h. The combined treatment was co-administered at the same time. All assays were performed in triplicate in at least three independent experiments. Synergy score: <−10 (antagonism, green), −10 to 10 (additivity, white) and >10 (synergism, red).

**Figure 6 ijms-24-00097-f006:**
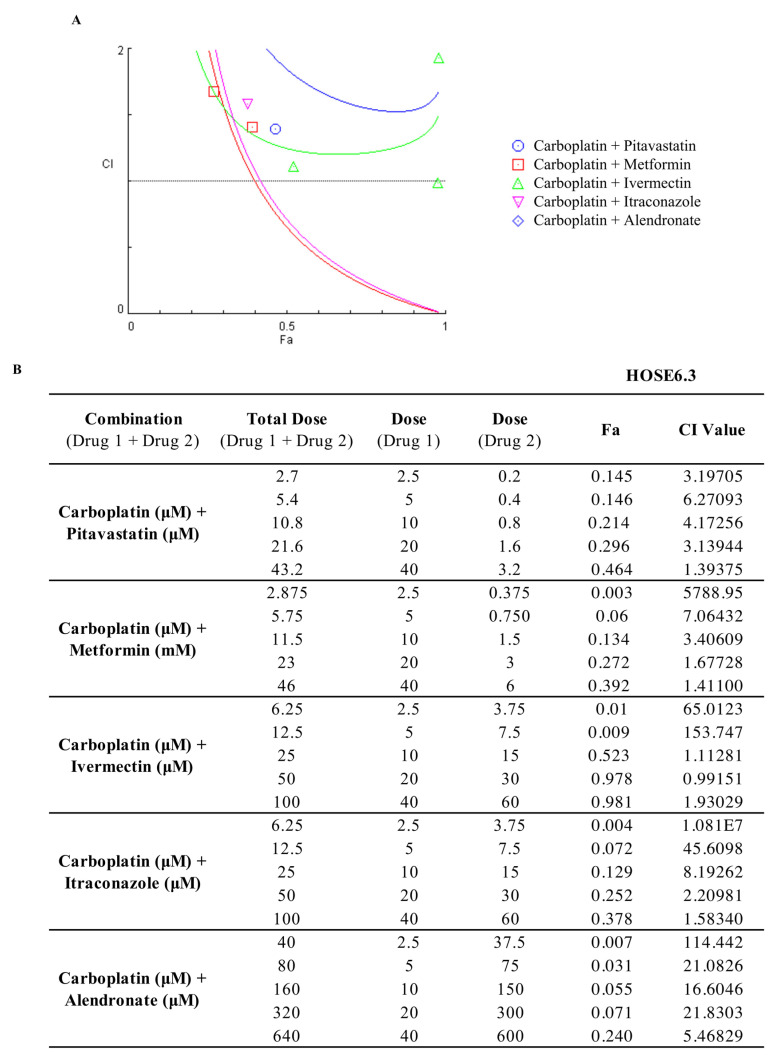
Combining Carboplatin with repurposed drugs has an antagonistic effect on HOSE6.3 cells. (**A**) Chou–Talalay method effect level (Fa)—Combinatory Index (CI) plot and (**B**) Fa values and respective combinatory index (CI) values showing drug synergism of HOSE6.3 cells, after exposure to a fixed dose ratio that correspond to 0.25, 0.5, 1, 2 and 4 fold the individual IC_50_ values each drug, e.g., Carboplatin combined with Pitavastatin, Metformin, Ivermectin, Itraconazole and Alendronate for 48 h. The combined treatment was co-administered at the same time. All assays were performed in triplicate in at least three independent experiments. CI was plotted on the y-axis as a function of Fa on the x-axis to evaluate drug synergism. CI: <1 (synergism), =1 (additivity) and >1 (antagonism).

**Figure 7 ijms-24-00097-f007:**
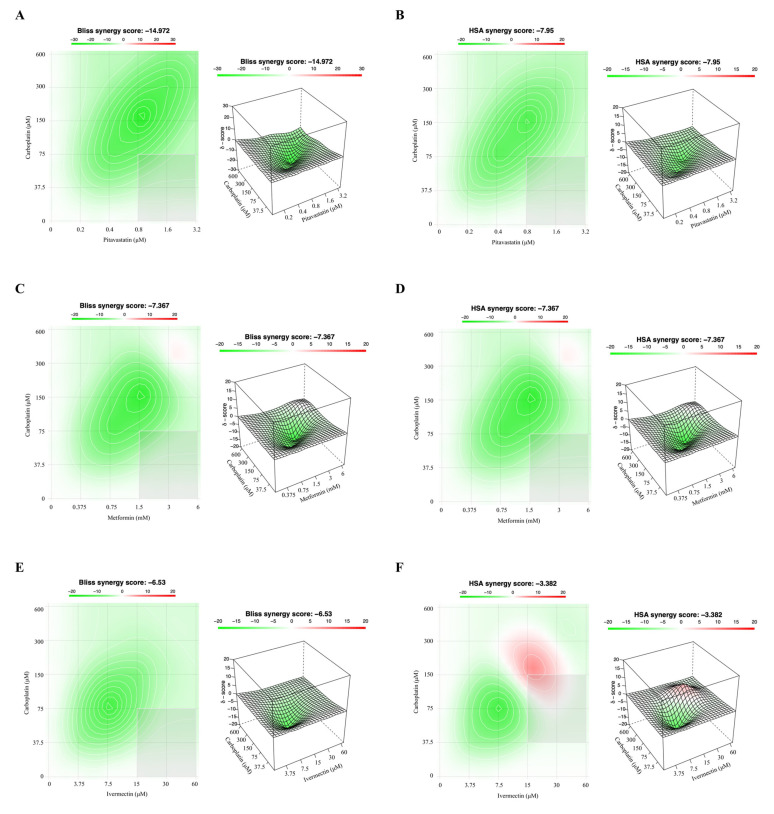
Combining Carboplatin with repurposed drugs has an antagonistic effect on HOSE6.3 cells. (**A**–**J**) Bliss Independence and High Single Agent (HSA) synergy 2D and 3D plots showing drug synergism of HOSE6.3 cells, after exposure to fixed dose ratios that correspond to 0.25, 0.5, 1, 2 and 4 fold the individual IC_50_ values of each drug, e.g., Carboplatin combined with (**A**,**B**) Pitavastatin, (**C**,**D**) Metformin, (**E**,**F**) Ivermectin, (**G**,**H**) Itraconazole and (**I**,**J**) Alendronate for 48 h. The combined treatment was co-administered at the same time. All assays were performed in triplicate in at least three independent experiments. Synergy score: <−10 (antagonism, green), −10 to 10 (additivity, white) and >10 (synergism, red).

## Data Availability

The data presented in this study are available in this article and Appendix A.

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
