# Peer review of "The Antineoplastic Effect of Carboplatin Is Potentiated by Combination with Pitavastatin or Metformin in a Chemoresistant High-Grade Serous Carcinoma Cell Line"

_ijms, 2022, doi:10.3390/ijms24010097_

Round 1

Reviewer 1 Report

The study focuses on the potentiation of carboplatin's anticancer impact when paired with either pitavastatin or metformin against Chemoresistant High-Grade Serous Carcinoma Cell Line. Overall, considering the importance of developing innovative anticancer therapy combinations and drug repurposing, the work is intriguing and adds to our understanding of cancer chemotherapy. As a result, the manuscript may be accepted for publication in the IJMS. I request that the authors proofread their work for minor grammatical faults and unfinished sentences. Additionally, some of the references lack the doi numbers. Please add them to make the references complete.

Author Response

We appreciate the time and dedication that the reviewer has given to provide this feedback about our manuscript. We have incorporated all reviewer’s recommendations:

  1. We have proofread the manuscript and changed some minor grammatical inaccuracies and unfinished sentences which are detailed in “track changes” in the new version of the manuscripts.
  2. We have added the DOI to the references that provide this information. Unfortunately, only in the reference 31 we could include this number. The other references (i.e., 38, 52, 64, 78, 90, 100, 101, 116, 124, 134 and 136) don´t have a DOI number available, so we could not add this information.

Reviewer 2 Report

The work by Nunes et al represents an interesting study for potential synergistic combinations of carboplatin for serous carcinoma. I recommend publishing after addressing the points below:

1. What is the history of combination therapy based on carboplatin (clinical and preclinical)? This needs to be clearly presented in the introduction. 

2. The quality of subfigures in Fig. 1 (specially text size) needs to be improved.

3. If additional cell lines can be tested, this would be advantageous. 

Author Response

We value the time and effort that the reviewer has dedicated providing feedback about our paper. We have integrated the suggestions made by the reviewer in our manuscript which are transcribed below.

1. We thank the reviewer for the valuable feedback. We agree and have included the following paragraph in the introduction (see “track changes”):

“The drug combinations have a long history being widely applied in the treatment of ovarian cancer (OC). The first described efficient combination was Platinum (i.e., Cisplatin or Carboplatin) with Cyclophosphamide [1]. In 1990´s, a trial performed by the Gynecologic Oncology Group showed that the combination of Cisplatin with Paclitaxel is more effective compared with the previous regimen [2]. Therefore, Paclitaxel was incorporated into the first-line therapy of OC, which brought significant improvement in the treatment, especially in the advanced disease [2]. An-other trial, in advanced OC (stage IIB-IV), demonstrated that Carboplatin-Paclitaxel as a good alternative to the Cisplatin-Paclitaxel combination, with similar efficiency and lower toxicity [3].

2. We appreciate the suggestion, and we have improved the quality of sub figures in Figure 1.

3. We acknowledge the reviewer for this suggestion. Indeed, we tested more cell line models with different treatment backgrounds, however, currently we don´t have capacity to include all this results in this paper. Moreover, our next research line is commited to perform drug efficacy testing’s using ex vivo models. Briefly, malignant ascites obtained from ovarian cancer patients by paracentesis will be used to develop patient-derived organoids models and these patient cells will be exposed to a panel of drugs to predict clinical responses. The most promising drugs will be selected to be administrated in patient in a personalized manner. Considering this, we have included the following paragraph in the conclusion (see the “track changes”):

“… To the best of our knowledge, our study is the first in showing that Pitavastatin and Metformin alone and combined with Carboplatin, increase the cytotoxic effect of this antineoplastic drug in a Carboplatin plus Paclitaxel chemoresistant HGSC model. …”. “The next step is evaluating the effect of these combinations in a brother panel of cell lines with different treatment backgrounds and expand our research to ex vivo models, since different patients present specific phenotypic characteristics, genotypic status and chemoresistance patterns. Indeed, the use of patient-derived organoids will allow the development of a drug sensitivity test capable to predict clinical responses to therapy.”
